# Frontier Materials for Adsorption of Antimony and Arsenic in Aqueous Environments: A Review

**DOI:** 10.3390/ijerph191710824

**Published:** 2022-08-30

**Authors:** Xiaohua Fu, Xinyu Song, Qingxing Zheng, Chang Liu, Kun Li, Qijin Luo, Jianyu Chen, Zhenxing Wang, Jian Luo

**Affiliations:** 1Ecological Environment Management and Assessment Center, Central South University of Forestry and Technology, Changsha 410004, China; 2South China Institute of Environmental Sciences, Ministry of Ecology and Environment, Guangzhou 510655, China; 3A.B Freeman School of Business, Tulane University, 6823 Saint Charles Ave, New Orleans, LA 70118, USA; 4Guangzhou Huacai Environmental Protection Technology Co., Ltd., Guangzhou 511480, China; 5School of Civil and Environmental Engineering, Georgia Institute of Technology, Atlanta, GA 30332, USA

**Keywords:** antimony, arsenic, frontier adsorption materials, heavy metal, water pollution treatment

## Abstract

As highly toxic and carcinogenic substances, antimony and arsenic often coexist and cause compound pollution. Heavy metal pollution in water significantly threatens human health and the ecological environment. This article elaborates on the sources and hazards of compound antimony and arsenic contamination and systematically discusses the research progress of treatment technology to remove antimony and arsenic in water. Due to the advantages of simple operation, high removal efficiency, low economic cost, and renewable solid and sustainable utilization, adsorption technology for removing antimony and arsenic from sewage stand out among many treatment technologies. The adsorption performance of adsorbent materials is the key to removing antimony and arsenic in water. Therefore, this article focused on summarizing frontier adsorption materials’ characteristics, adsorption mechanism, and performance, including MOFs, COFs, graphene, and biomass materials. Then, the research and application progress of antimony and arsenic removal by frontier materials were described. The adsorption effects of various frontier adsorption materials were objectively analyzed and comparatively evaluated. Finally, the characteristics, advantages, and disadvantages of various frontier adsorption materials in removing antimony and arsenic from water were summarized to provide ideas for improving and innovating adsorption materials for water pollution treatment.

## 1. Introduction

Since the 20th century, there has been an increase in awareness of heavy metal contamination, which has resulted in several industrial accidents and environmental issues. Antimony and arsenic pollution occurrences have become more frequent as a result of the progressive increase in the usage of antimony (Sb) and arsenic (As) in many nations worldwide in recent decades [1], which have influenced both the environment and human health negatively. Antimony and arsenic share the same primary group (Group VA) in the periodic table of elements and have comparable chemical characteristics [2,3]. They have both metallic and nonmetallic physical and chemical characteristics. Antimony and arsenic are primarily found in both III and V oxidation forms in the environment. They typically exist in the states of Sb(V) and As(V) in aerobic environments, but in environments with low oxygen levels, they exist in the states of Sb(III) and As(III) [4,5]. The trivalent forms of antimony and arsenic are significantly more toxic than pentavalent antimony and arsenic [6,7].

Part of the cause of Sb and As pollution comes from nature itself, and most of it comes from human activities, such as mining activities, the use of pesticides and insecticides, and the production of electronic industries [1]. Combustion of coal and fossil fuels also volatilizes Sb and As, and most metal sulfides and coal contain Sb. In particular, coal with high As content generally has relatively high Sb content. Coexistence of Sb and As pollution is widespread [8]. The World Health Organization (WHO) stipulates that the maximum allowable concentration of Sb in drinking water is 20 µg/L, while China stipulates that the maximum allowable concentration in drinking water is 5 µg/L [9]. The World Health Organization and the Ministry of Health of China revised the concentration of arsenic in drinking water from 50 µg/L to 10 µg/L in 1993 and 2007, respectively [10].

Currently, the pollution treatment technologies of antimony and arsenic in the water environment mainly include coagulation/flocculation, membrane technology, oxidation methods, electrochemical methods, adsorption methods, etc. [7,11,12]. Compared with other technologies, adsorption technology has the advantages of lower cost, simpler process, and regenerable adsorption materials, which has become a research hotspot. For adsorption technology, the choice of adsorbent greatly affects the adsorption effect. Many adsorbents are used in the adsorption of heavy metals in the water environment, but there are many types and different adsorption effects. Many frontier materials have attracted increasing attention from researchers at home and abroad due to their superior adsorption properties. Therefore, this article focuses on summarizing the characteristics, adsorption mechanism, and performance of the frontier adsorption materials reported so far, describing their research and application progress in the removal of antimony and arsenic in the water environment, and conducting an objective analysis and comparative evaluation of the adsorption effects of a variety of frontier adsorption materials.

This review is divided into the following parts: (1) Chemical properties, hazards, source distribution, and forms of existence in different environments of Sb and As; (2) Sb and As pollution control technology; (3) Characteristics, mechanism, and application of frontier adsorption materials; (4) Research progress on the adsorption of Sb and As by frontier materials; (5) Summary and prospects.

## 2. Chemical Properties, Hazards, and Sources of Antimony and Arsenic

### 2.1. Chemical Properties of Antimony and Arsenic

The atomic mass and the atomic number of antimony are 121.76 and 51, respectively. It belongs to the periodic table’s metalloid element group VA. Sb(III) mainly exists in the form of Sb(OH)_3_/SbO(OH) or HSbO_2_ in the pH range from 1 to 11 [6]. Under stable acid conditions, the existing form of Sb(III) changes to mainly SbO^+^ or Sb(OH)_2_^+^ positive ions. Under strong alkaline conditions, the existing form of Sb(III) is mainly negatively charged ions SbO_2_^−^ or Sb(OH)_4_^−^ [6].

According to the periodic table, arsenic has an atomic mass of 74.92 and an atomic number of 33. It belongs to the VA group of metalloid elements. Dimethyl arsenic acid, monomethyl arsenic acid, and other types of organic arsenic are now available [13]. Generally, organic arsenic is less toxic than inorganic arsenic [14]. In the water environment, As mainly exists in the inorganic form of As(III) or As(V) [7,15]. Due to its stable condition, As(V) has a larger concentration than As(III) in natural water, with a pH value between four and eight. When the pH is less than nine, As is primarily presented in anaerobic groundwater as As(III) [1].

### 2.2. The Harm of Antimony and Arsenic to Human Health

Sb and As not only pollute the ecological environment but also bring a massive threat to human health. The international IARC team believes Sb(III) may cause cancer [11]. Sb(V) and cancer may be related, but this has not yet been confirmed. Sb poisoning can be split into roughly two categories. The first is the clinical signs of acute poisoning, which include vomiting, hematuria, and stomach pain. The other is chronic poisoning caused by chronic bronchitis, emphysema, pleural adhesions, and early tuberculosis due to long−term exposure to low concentrations of Sb [11,16].

As may enter the body of a human in two main ways. One is to ingest water that has been tainted with As directly. Around the world, drinking water issues brought on by arsenic poisoning have put more than 200 million people’s health at peril [17]. Consuming meals high in As is another route for As to enter the body. According to the World Health Organization (WHO), As will be harmful to human health if people consume it over an extended period and the cumulative concentration is more than 50 g/L. It may cause a range of diseases, including skin cancer, visceral cancer (bladder cancer, kidney cancer, lung cancer), diabetes, hypertension, and diseases of the reproductive system by destroying the integrity of human cells and genetic material [7,15].

### 2.3. Pollution Sources of Antimony and Arsenic

The pollution sources of Sb and As are shown in Figure 1, which can be divided into natural and artificial sources. Natural sources of Sb and As pollution include volcanic eruptions, geothermal activities, forest fires, and sandstorms [1,18].

Anthropogenic sources are also the main contributors to Sb and As pollution compared to natural sources. The primary manufactured sources are mining, burning fossil fuels, refining nonferrous metals, manufacturing pesticides, preserving wood, and the electronics sector, among others [6,7,11]. For example, coal combustion and metal smelting produce Sb and As [8,19]. With the discharge of industrial effluent or the erosive action of rains, chemicals such as Sb and As created during combustion or smelting will infiltrate the soil and water bodies. A small proportion of nonrecyclable antimony products and wastes containing antimony that are produced throughout the production process are the primary contributors to Sb pollution in the environment. Arsenic is widely used as an insecticide and pesticide. After entering the soil, it can be leached out by rain and enters the water body, causing water pollution.

### 2.4. Distribution of Antimony and Arsenic in Water

The environmental factors affecting the existence of antimony and arsenic in water mainly include pH and redox conditions. Sb usually has four oxidation states in water, including Sb(−III), Sb(0), Sb(III), and Sb(V). In aerobic surface water and anaerobic groundwater, the leading valence states of Sb are respectively Sb(III) and Sb(V) [20]. Arsenic, such as antimony, is often present in the form of soluble arsenic, namely As(III) and As(V). Under oxidizing conditions, arsenic exists mainly in pentavalent arsenic, including H_3_AsO_4_, H_2_AsO_4_^−^, HAsO_4_^2−^, and AsO_4_^3−^. Under anoxic conditions, trivalent arsenic is its main form in water H_3_AsO_3_, H_2_AsO_3_^−^, HAsO_3_^2−^, and AsO_3_^3−^ [21]. Sb(V) and As(V) exist stably in an oxygen−containing environment, while in anoxic environments, Sb(III) and As(III) exist stably in groundwater or pore water.

In addition to redox conditions, pH is also a key factor affecting the presence of antimony and arsenic. In the range of pH = 2.7–10.4, Sb(III) mainly exists in the form of neutral Sb(OH)_3_; while when the pH value is greater than 2.7, Sb(V) mainly exists in the form of negatively charged H_2_SbO_4_^−^ or Sb(OH)_6_^−^ [22]. According to the effect of pH on the form of As in previous studies, when the pH value is 5.0 to 8.0, the primary forms of AS(V) are H_2_AsO_4_^−^ and HasO_4_^2−^, while As(III) is H_3_AsO_3_. Under oxidative conditions, HasO_4_^2−^ is the main form under high pH conditions, and the contents of H_3_AsO_4_ and AsO_4_^2−^ are higher under extremely acidic and alkaline conditions. Under the condition of low pH (<6.9), H_2_AsO_4_ is dominant. This means that As(III) is still a neutral molecule in natural water [21].

#### 2.4.1. Surface Water

In surface water, antimony exists mainly in the form of Sb(V) and organic antimony, including H_2_SbO_4_^−^ and Sb(OH)_6_^−^ as well as monomethyl antimonic acid and dimethyl antimonic acid. Arsenic is mainly pentavalent arsenic forms, including H_3_AsO_4_, H_2_AsO_4_^−^, HasO_4_^2−^, and AsO_4_^3−^. Surface water, groundwater/drinking water, and seawater include Sb and As. Sb primarily enters surface waters through soil or rain from the atmosphere. The average Sb concentration in rivers around the world is about 1 g/L [23]. The European Union tested 807 surface water samples and found that the Sb concentration was around 0.002–2.910 µg/L [11]. The baseline concentration of As in rivers was specified in the range of 0.1–2.0 µg/L [1]. The concentration of As in lakes is usually regulated to be less than 1 µg/L or close to 1–2 µg/L [1]. There have been instances of excessive As concentrations in lakes throughout the world. For instance, the water’s arsenic content ranged from 1.39 to 6.65 g/L in Taihu Lake to 3.08 to 10.48 g/L in Dianchi Lake [24].

#### 2.4.2. Groundwater

Sb(III) and As(III) are the main forms in groundwater and sediments, including Sb(OH)_3_ and H_3_AsO_3_, H_2_AsO_3_^−^, HAsO_3_^2^^−^, and AsO_3_^3^^−^ [20]. With the increasingly severe antimony pollution, drinking water quality problems frequently occur. The concentration of Sb in geothermal groundwater is generally 0.06–26 µg/L [25]. The concentration of Sb detected in 476 groundwater samples in Norway was 0.002–8 µg/L [26]. In most countries, the limited concentration of arsenic in groundwater followed the international standard of 10 µg/L. In countries more severely affected by arsenic pollution, such as Bangladesh and other Asian countries, the maximum allowable level of arsenic in drinking water follows 50 µg/L [27].

#### 2.4.3. Sea Water

The concentration of Sb in the ocean was approximately 0.2 µg/L, and there was no accumulation of Sb in the deep−water marine environment [11]. For example, the concentration of Sb was approximately 0.3–0.82 µg/L in the North Sea near the Belgian coast and 0.26 µg/L in the Irish Sea [18]. The concentration of As in seawater was generally less than 2 µg/L [28]. For example, the concentration of As ranged from 1.0–1.8 µg/L along the deep Pacific and Atlantic coasts, 0.7–1.8 µg/L off the coast of Malaysia, and 0.5–3.7 µg/L off the coast of Spain [29].

## 3. Antimony and Arsenic Pollution Treatment Technology

### 3.1. Coagulation/Flocculation Method

Coagulation and flocculation have several benefits, including cheap operating costs, use easily, suitability for large−scale water treatment, and efficient removal of heavy metals over a wide pH range [30]. Using a coagulant to interfere with the charge on the surface of colloidal particles to neutralize the charge, which eliminates the repulsive force between the particles and causes the particles to mesh with one another, is the mechanism of coagulation/flocculation to remove heavy metals.

Coagulation/flocculation generally uses iron flocculants to remove Sb, and the coagulation−flocculation−sedimentation (CFS) removal process is shown in Figure 2. The clearance rate of Sb(V) can reach 98% when the iron flocculant has a specific dosage and when the pH ranges between 4.5 and 5.5. Sb(III) requires less iron flocculant than Sb(V), which may be efficiently removed over a more comprehensive pH range of 4–10 [31,32].

On the one hand, some positively charged coagulants lower the negative charge of the colloid when arsenic is removed using coagulation procedures, which leads to the production of bigger particles. On the other hand, some positively charged coagulants lower the negative charge of the colloid when arsenic is removed using coagulation procedures, which leads to the production of bigger particles [30]. At present, the commonly used and effective coagulants for removing As from water are aluminum salts and iron salts [7], which are more economical and practical.

### 3.2. Ion Exchange Method

The ion exchange method works with the reversible equal interchange of ions in diluted solution with those in solid ion exchangers [33]. Both adsorption and the ion exchange process used to remove heavy metals work by absorbing solutes from the solution. Ion exchange benefits from excellent efficiency and little sludge [6].

Some researchers discovered that amino phosphonic acid resin could successfully remove Sb(III) and Sb(V) in the copper electrolyte, with Sb(III) removal efficiency being significantly greater than Sb(V) [34]. Inorganic ion exchangers, including magnesium−aluminum, copper–aluminum, and other layered double hydroxides, are also employed for ion exchange to remove Sb in addition to organic ion exchangers. The reaction mechanism is the exchange of Sb(OH)_6_^−^ and NO_3_^−^ in the interlayer of Mg−Al and Cu−Al LDH [35].

Ion exchange removal of As usually uses a strong base anion exchange resin. For example, Amberlite XAD−7 resin loaded with titanium dioxide is used to explore its adsorption performance for As(III) and As(V). Consequently, at pH 5–10 and pH 1–5, As(III) and As(V) showed sound adsorption effects. This resin can absorb As(III) more quickly and with a higher capacity than As(V) can [36].

### 3.3. Membrane Technology

Traditional treatment methods such as membrane technology have the benefits of high removal effectiveness, low operational energy consumption, and specific operating conditions [11]. Pore size can be divided into four categories: microfiltration (MF), ultrafiltration (UF), nanofiltration (NF), and reverse osmosis (RO) [16]. The mechanism of this technology is the selective permeation of membranes [11].

Metal ions travel through the membrane to chelating groups quicker than water molecules when membrane technology is employed to extract Sb, and wastewater containing Sb passes through the membrane. Reverse osmosis can therefore be used to remove Sb(V) and produce a superior result than Sb(III) regardless of the pH value [11]. Sb(V) can also be removed by the strengthening ultrafiltration membrane method, and its process is simple to operate [37].

Although all four membrane technologies can effectively remove As, the best results are nanofiltration (NF) and reverse osmosis (RO) [15]. As(III) is more difficult to remove, and As(V) has a better removal rate. Therefore, some studies have used reverse osmosis to remove As(V) in water and established a model to optimize the system treatment process [38]. Kang et al. investigated the effects of variables such as pH value on the removal efficiency of Sb and As by the RO membrane because they thought the RO membrane could remove Sb and As simultaneously [39]. The findings demonstrate that Sb(V) and As(V) have substantially greater removal efficiencies than Sb(III) and As(III).

### 3.4. Electrochemical Method

An electrochemical method is a form of electrocoagulation (EC), which dissolves the soluble anode by applying an electric current between iron electrodes [7]. The primary functions of electrochemical technology include micro electrolysis, oxidation, flocculation, and coagulation. Through particle bridging and coprecipitation, the sacrificial anode creates several metal hydroxides or coagulants that can efficiently adsorb contaminants [11]. It is an increasingly popular treatment of high−concentration wastewater, such as electrolytic refining, dye factory waste liquid, peanut plant wastewater, and chemical power supply.

Underpotential deposition (UPD) is frequently used in electrochemical processes to remove Sb from water. This technique uses potential sweeps and steps in an electrochemical setting. Relevant professionals have investigated copper electrolytic refining of electrodeposition technology and waste battery solutions [40]. After electrocoagulation with a Fe−Al electrode, the removal efficiency of Sb exceeded 99% [41]. Electrocoagulation is an alternative technology for coagulation/flocculation to remove As. According to Lakshmanan et al., As(III) may be successfully removed when copper−copper and zinc−zinc electrodes are used [42]. Metal−air fuel cell electrocoagulation (MAFCEC) has been suggested as a potential solution to the problems with the conventional EC procedure in recent years [43]. It is more energy−saving, environmentally friendly, and efficient. Therefore, it has broader prospects in the field of electrocoagulation removal.

### 3.5. Phytoremediation Technology

Phytoremediation is a low−cost, ecological, and environmentally friendly heavy metal treatment technology [15]. The main mechanism of phytoremediation to remove heavy metals is through plant extraction, stabilization, volatilization, and rhizosphere filtration [44].

The application of phytoremediation technology in As pollution is more prominent. Researchers have found that ferns have strong As removal capabilities and can be widely used. For instance, Nazir et al. investigated the phytoremediation capacity of Cd, As, and Hg absorption in water hyacinth [45]. The findings demonstrate that it has the best Cd adsorption impact and several As and Hg adsorption capabilities. Ferns can also eliminate both Sb and As at once. For example, Pteris cretica L is a plant that can jointly remove high concentrations of Sb and As in water, and its adsorption capacity can reach 1.677 mg/g and 1.517 mg/g, respectively [46]. The applications of different pollution treatment technologies in removing Sb and As are listed in Table 1.

### 3.6. Adsorption Method

Adsorption is a method for removing chemicals from gas or liquid solutions by using solids as a medium [7]. The major driving forces behind the entire process are the van der Waals force and electrostatic attraction between adsorbent molecules and surface atoms [43]. The adsorption properties of different adsorbents for Sb and As are listed in Table 2.

With the continuous deepening of antimony and arsenic adsorption research, the adsorbents that can remove antimony and arsenic are becoming increasingly diversified. Among them, the more common adsorbents with good adsorption effects are activated alumina [6,55], activated carbon [56,57], manganese dioxide [58,59], iron hydroxide [60,61], zeolite [62,63], clay [64,65], zero−valent iron [66], biomass material [67] and so on. However, several academics have demonstrated recently that iron−based adsorbents, which are inexpensive and simple to recover, have a stronger adsorption impact on Sb and As than other conventional adsorbents. As a result, eliminating Sb and As from iron−based materials has been a popular area of research for many academics [68,69,70]. In addition to iron−based materials, many frontier materials are also used to remove Sb and As in aqueous environments, such as MOFs [71], COFs [72], graphene [73], graphene oxide [74], and hydrogel composites [75]. When comparing the data in Table 2, it can be found that the adsorption effect of frontier materials such as MOFs, COFs, and many iron−based materials for Sb and As is much higher than that of some traditional materials.

At present, the existing research is limited to the single solute adsorption of Sb and As, and there is a lack of research on the synergistic removal of Sb and As compound pollution. However, many studies have proven that the adsorbent for the removal of Sb and As has a good effect, and the efficiency is greater than the removal of a single solute. Some scholars have compared the effects of Sb and As single adsorption and mixed synergistic adsorption. For example, activated alumina, a commonly used traditional adsorbent, had a single adsorption capacity of 7.72 mg/g for antimony. In comparison, the maximum adsorption capacity of antimony in an antimony−arsenic mixed solution was 11.6 mg/g, 1.5 times that of single adsorption. The maximum adsorption capacity of As(III) in the antimony−arsenic mixed solution also increased from 5.84 mg/g for single adsorption to 7.38 mg/g, an increase of nearly 1.3 times. It can be inferred that most adsorbents had a higher synergistic adsorption effect on antimony and arsenic than the single adsorption of antimony and arsenic [76]. However, there are few studies on the coremoval of Sb and As, and the in−depth exploration of its adsorption mechanism is lacking.

### 3.7. Advantages and Disadvantages of Various Technologies

The above techniques have been widely used to reduce Sb and As contamination, and various techniques have certain advantages and limitations in removing Sb and As. When using flocculation/coagulation technology, iron coagulants are often used to remove Sb and As, which are low in cost, have a large allowable pH value range, and are easy to operate. However, if there is a large amount of flocculation/coagulation by−products, a large amount of toxic sludge containing Sb and As will be produced, causing secondary pollution [11]; The advantages of the ion exchange method are high efficiency and less sludge; however, the process of removing Sb and As is hindered by other competing ions, such as Cl^−^ and HCO_3_^−^, which affects the adsorption effect [77]; For membrane technology, because of its simple operation, high removal efficiency, low energy consumption, and small footprint, it is widely used in many chemical technologies, especially for water treatment. However, membrane filtration can easily cause membrane blockage, high investment and maintenance costs, and is challenging to recycle [11]; Electrochemical method is an efficient sewage treatment method commonly used to treat high−concentration wastewater. However, its adsorption effect is highly dependent on pH value, electrode material, and time setting and has disadvantages such as high cost and large sludge discharge [6]; Phytoremediation has the advantages of the low cost of raw materials and high selectivity. However, this technology takes a long time to remove Sb and As, and the removal effect on heavily polluted sites is not outstanding [4,10,13]. Adsorption methods generally have the advantages of simple operation, good Sb and As removal effects, low cost, high efficiency, and strong regeneration ability [15]. These advantages make it stand out from traditional Sb and As pollution treatment technologies. However, the adsorption method also has certain limitations, and its adsorption effect mainly depends on two aspects. The first is the adsorbent’s composition, while the second is its type. Temperature, pH, and other interfering ions all have an easy way of affecting the adsorption action. The adsorption of antimony and arsenic is thus greatly influenced by the chemical characteristics of the adsorbent [11,78].
ijerph-19-10824-t002_Table 2Table 2Adsorption capacity and other parameters of different adsorbents for removing antimony and arsenic.AdsorbentHeavy MetalInitial Concentration (mg/L)Adsorbent Dose (g/L)Adsorption Temperature (°C)Optimum pHAdsorption Capacity(mg/g)ReferencesMNP@hematiteSb(III)0.110.1254.136.70[79]Erzurum claySb(III)20010251.59.20[64]DiatomiteSb(III)10420635.20[80]Mercapto−functionalized hybrid sorbentSb(III)515.35255108.80[81]BentoniteSb(III)/Sb(V)1252560.56/0.50[82]ZCNSb(III)/Sb(V)100–500125770.83/57.17[83]RGOSb(III)/Sb(V)0–3001 × 10^−3^256168.59/206.72[73]MIL−101(Fe)Sb(III)/Sb(V)5–2500.5−6151.80/472.80[71]Iron oxide coated cement (IOCC)As(III)0.7–13.5301570.73[84]Granular ferric hydroxide (GFH)As(V)0.010.25206.51.10[60]Synthetic zeolite H−MFI−24As(V)10–1502206.535.80[85]Natural clayAs(V)20225386.86[86]Activated AluminaAs(III)/As(V)0.79–4.9/2.85–11.51256.9/5.23.50/15.90[87]Iron−modified activated carbonAs(III)/As(V)20–22−−638.80/51.30[57]MAF−RGOAs(III)/As(V)0.1–1000.2−7402.00/339.00[88]Fe−Co−MOF−74As(III)/As(V)1–2500.5253/7266.00/292.00[89]

## 4. Introduction of Frontier Adsorption Materials

Thus far, there are an increasing number of types of adsorbents, as shown in Figure 3. In recent years, porous organic materials have become a new generation of high−efficiency adsorbents, which have the advantages of low synthesis cost, large specific surface area, and adjustable pore size. These porous organic materials include metal−organic frameworks (MOFs), covalent organic frameworks (COFs), and hydrogen−bonded organic frameworks (HOFs). Graphene evolved from the graphite flakes discovered by Geim and Novoselov [90]. As a derivative of graphene, graphene oxide (GO) can overcome the shortcomings of graphene hydrophobicity, so it is widely used for heavy metal adsorption in wastewater treatment. As a 2D material similar to graphene, MXenes are also used in wastewater treatment.

### 4.1. Metal−Organic Frameworks (MOFs)

Metal−organic frameworks (MOFs) are a form of crystalline coordination polymers made up of transition metal clusters or ions and polyhedral or dual−system organic ligands [91]. MOFs also provide benefits over currently used adsorbents that other adsorbents cannot match. MOFs can be produced on a wide scale because their synthesis is inexpensive and relatively straightforward. Regarding structure, MOFs have a much larger surface area than conventional materials, such as zeolite, permanent porosity, and adjustable pore size [91].

In the following 20 years, the types of MOFs were continuously enriched. At present, the types of MOFs are mainly the MIL series, ZIF series, UiO series, etc. [92]. In addition, the magnetic MOF composite material has a better adsorption effect for heavy metals. For example, Karimi and others have adopted a new green strategy to successfully prepare a magnetic metal−organic framework nanocomposite material (Fe_3_O_4_−NHSO_3_H@HKUST−1), which has an adsorption capacity of approximately 384.6 mg/g for lead ions in water [93]. The adsorption of Pb^2+^ by Fe_3_O_4_−NHSO_3_H@HKUST−1 is a spontaneous and heat−absorbing process, the main mechanism of which is the coordination of Pb^2+^ with —NH_2_ on the surface of this adsorbent. In recent years, compounds such as porphyrins, which have a cyclic structure and can efficiently remove heavy metal ions, have been successfully used as key materials in the synthesis of MOFs. For example, Hasankola et al. have synthesized a zirconium−based MOFs material called PCN−221 using a solvothermal method with 5,10,15,20−tetrakis (4−carboxyphenyl) porphyrin (H2TCPP) as a linker [94]. The porphyrin ligand in the structure has a nitrogen−functionalized group with strong electron−donor properties, which can effectively adsorb Hg^2+^ under ultrasonic water bath conditions and further enhance the Hg^2+^ removal performance through the formation of metal−nitrogen coordination bonds. The maximum adsorption capacity of the adsorbent was 233 mg/g in a short time at pH−neutral conditions. However, there are few kinds of research on magnetic MOF materials, and they have not been put into large−scale preparation.

### 4.2. Organic Framework Material

#### 4.2.1. Covalent Organic Frameworks (COFs)

The covalent organic framework is a novel type of organized active crystal porous polymer that connects organic monomers with light elements such as C, O, N, and B through strong covalent connections [95]. COFs are derived from MOFs. They resemble MOFs because they have highly organized pore diameters, adaptable, flexible, and varied structures, large surface areas, many functional sites, and chemical stability [96,97]. Unlike MOFs, COFs have a more ordered channel structure, lower density, and higher thermal and chemical stability.

In 2005, Yaghi and others successfully synthesized two−dimensional COFs for the first time [98], and COFs have been widely used in many fields, such as photoelectric sensors and energy storage. COFs have started to be utilized to remove heavy metals in recent years. Coordination bonds/chelation effects, electrostatic interactions, hydrogen bonds, and ion exchange are the key methods by which COFs remove heavy metals [99]. Another COF composite material, COF−LZU8, has a removal efficacy of up to 98% and can achieve Hg^2+^ in the pH range of 3 to 13 [100]. The high adsorption rate of COF−LZU8 for Hg^2+^ was mainly due to the high stability of the hydrazone linkages, as well as the dense distribution of thioether groups and linear channels in this COF material. In order to focus on improving the adsorption capacity and reaction rate of COFs for heavy metal removal, a covalent organic skeleton (DMTD−COF−SH) using 2,5−dimercapto−1,3,4− thiadiazole (DMTD) and 1,2−ethanedithiol doubly modified COF−V was synthesized in a recent study of COFs [101]. The ordered mesopores of the COF combined with the highly dense distribution of sulfur and nitrogen groups attached to the COF surface have a synergistic effect, resulting in a COF with large porosity and an abundance of accessible chelation sites. As a result, DMTD−COF−SH exhibited high adsorption capacity, fast adsorption rates, and strong stability for a variety of coexisting trace heavy metals such as lead, mercury, cadmium, chromium, and copper in tap water. The most significant adsorption effect of this COF was for lead, with a maximum adsorption capacity of 14.22 mg/g.

#### 4.2.2. Hydrogen−Bonded Organic Frameworks (HOFs)

Polymers with crystalline pores are called hydrogen−bonded organic frameworks (HOFs). It is a group of compact organic molecules made up of hydrogen bonds, stacking forces, and van der Waals interactions amongst light elements such as C, H, O, N, and B [102]. HOF materials have the advantages of both MOFs and COFs and are relatively simple to manufacture, gradually becoming an emerging type of porous organic framework (POP). After the 1990s, Wuest and other scholars discovered a large number of HOFs and began to devote themselves to the design and synthesis of HOFs [103]. Based on Wuest’s research, He et al. developed the first permanent porous, microporous organic framework HOFs with DAT−containing tetrahedral organic small molecule building units [104]. In contrast to the common application of HOFs on pure organic linkers or building blocks, Bao et al. argued that the addition of metal complexes or metal structures to HOFs facilitated the generation of unique pore structures, topologies, and functions and attempted to develop novel HOF materials using metal complexes as building blocks [105]. They constructed another HOF material (HOF−21) with excellent C_2_H_2_/C_2_H_4_ separation performance. HOF−21 was assembled mainly through [Cu_2_(ade)_4_] with co−loaded SiF_6_^2^^−^ anions, which exhibited extremely high stability and regenerative properties. The treatment of heavy metals in wastewater has not been used, and research on relevant topics has decreased.

### 4.3. Graphene

A single layer of carbon atoms arranged in the form of a two−dimensional honeycomb lattice makes up graphene. It is very elastic and thermally and electrically conductive [106]. With the benefits of a high specific surface area, strong hydrophilicity, and low toxicity, graphene oxide (GO), a graphene derivative, may efficiently remove heavy metals from wastewater while overcoming the drawbacks of graphene’s hydrophobicity [107]. GO is decorated with numerous reactive functional groups such as —OH, —COOH, —C = O, and other hydrophilic groups. In recent years, researchers utilized coprecipitation to produce magnetic graphene oxide (MGO) and used it as an adsorbent to adsorb heavy metal ions from water. MGO has good adsorption properties and regeneration due to the effects of metal chelation and electrostatic attraction. Its maximum adsorption capacities for Cr^3+^, Pb^2+^, Cu^2+^, Zn^2+^ and Ni^2+^ are 24.330 mg/g, 200.000 mg/g, 62.893 mg/g, 63.694 mg/g and 51.020 mg/g, respectively [108]. The synthesis of modified graphene and its composite materials and its application in the field of heavy metal wastewater treatment has become a research hotspot, but research on the adsorption mechanism still needs to be strengthened.

### 4.4. MXenes

MXenes are two−dimensional materials similar to graphene and are mainly obtained by extracting the A−site element in the Max phase from a mixture of HF acid and fluoride [109,110]. MXenes have excellent mechanical strength, high metal conductivity, excellent ion adsorption capacity, and unique topological structure [111]. In 2011, Michel W. Barsawm and Yury Gogotsi discovered new material, MXenes [112]. Peng et al. prepared alk−MXenes, which have activated hydroxyl groups and can effectively adsorb Pb(II) from water, and the equilibrium state was reached within 2 min [113]. Fard et al. prepared Ti_3_C_2_T_x_ nanosheets, which had excellent effects in absorbing Ba(II) in wastewater, and the absorption of Ba(II) ions by Ti_3_C_2_T_x_ was not interfered with by other competing ions [114]. To further enhance the performance of MXenes in removing heavy metals, many scholars have turned to the study of MXenes composites. For example, Gan et al. used levodopa (DOPA), an amino acid, as a modifier to modify the surface of MXenes to make MXenes composites called Ti_3_C_2_T_X_−PDOPA for effective removal of Cu^2+^ from water bodies [115]. Compared with the raw material Ti_3_C_2_T_X_, Ti_3_C_2_T_X_−PDOPA introduced many carboxyl groups, which led to a particular improvement in the adsorption performance of Cu^2+^, with a maximum adsorption capacity of 65.126 mg/g. In conclusion, MXene− and MXene−based nanomaterials are some adsorbents with great potential in adsorbing Sb and As in wastewater.

### 4.5. Other Adsorbents

Other frontier adsorbents, such as iron−based materials and composite hydrogels, have also been widely used to remove Sb and As. Iron−based materials include iron oxide, zero−valent iron, iron−based bimetallic oxides, and other Fe−loaded adsorbents. Compared with other adsorbents, iron−based adsorbents have incomparable advantages, such as strong hydrophilicity, large adsorption capacity, low cost, easy recovery, environmental friendliness, etc. [68]. It is worth mentioning that iron−based adsorbents generally outperform other metal adsorbents in function. For example, some magnetic iron oxides are easy to recover and reuse [116], which lays the foundation for the practical application of antimony and arsenic adsorbents.

Hydrogels are three−dimensional cross−linked polymeric materials synthesized through a simple chemical reaction between one or more monomers and a cross−linking agent. It can adapt to environments with different pH values, temperatures, and ion concentrations and can expand or shrink [117]. Hydrogels are widely used in the delivery of genetically engineered drugs [118], adsorption [119], sensors [120], and many other fields. Specific functional groups in hydrogels, such as −OH, −NH_2_, −COOH, −CONH_2_, are beneficial for removing metal ions from water [121]. It is worth mentioning that the hydrogels can be modified or composited with other materials to deal with different heavy metals or metalloids according to actual needs [122], and the adsorption effect of the synthesized composite hydrogel is better than that of the natural hydrogel [75]. In addition, some researchers have also explored the cyclic adsorption capacity of hydrogel composites in the process of desorption of pollutants [119], demonstrating their good reusability and recyclability. Therefore, hydrogel composites are potential frontier adsorbents for the coremoval of antimony and arsenic. The comparison of frontier and traditional adsorption materials is shown in Table 3.

## 5. Application of Frontier Adsorption Materials to Remove Antimony and Arsenic

### 5.1. MOFs Removal of Antimony and Arsenic

In recent years, extensive research has been conducted on the adsorption of Sb and As. The practice has proven that MOFs are a class of frontier materials that can effectively remove Sb and As and achieve good results.

The study of MOF material adsorption of Sb has also become a hot research field in recent years. Li et al. screened seven zirconium−based metal−organic frameworks, namely, Zr−MOFs, and used several different organic linkers, such as −NH_2_ and −OH. The most effective Zr−MOFs for adsorbing Sb were NU−1000, Sb(III), and Sb(V), which exhibited adsorption capacities of 136.97 mg/g and 287.88 mg/g, respectively [138]. Amino−modified zirconium metal−organic frameworks (UiO−66(NH_2_)) have also been shown to remove Sb. X−ray photoelectron spectroscopy (XPS) and FTIR analysis indicated that the material’s amino groups and Zr−O bonds play critical roles in the removal of Sb from water [139]. Cheng et al. systematically studied the effect of the iron−based metal framework Fe−MIL−88B on the removal of Sb(III) and Sb(V) [140]. The mechanism of Sb adsorption onto Fe−MIL−88B is shown in Figure 4. It is demonstrated that the production of HFO (hydrated oxide) and the coordination bond in Fe−MIL−88B work together to facilitate the adsorption of Sb. The findings demonstrate that at pH values of 10 and 6, respectively, Fe−MIL−88B has the best adsorption effects on Sb(III) and Sb(V), and its maximum adsorption capacities are 566.1 mg/g and 318.9 mg/g, respectively.

MOF materials have also been used to adsorb arsenic. In 2018, WU et al. used a hydrothermal method to synthesize iron−based MOF−MIL−88A for the adsorption of As(V) [141]. The results show that the adsorption capacity of arsenic can reach 145 mg g^−1^ and the adsorption speed is fast. Li et al. synthesized MIL−53(Al) for the adsorption of As(V), and hydrogen bonding and electrostatic interaction were used to complete the procedure [142]. The most extraordinary adsorption capacity was 105.6 mg/g in 1 h when pH = 8, producing the best adsorption effects. Some scholars have successfully prepared the zirconium metal−organic framework UiO−66, and the main mechanism of its adsorption of As(V) is shown in Figure 5. In contrast to earlier research, UiO−66 has a broader pH range for adsorption and can efficiently adsorb As in the pH range of 1 to 10. The largest amount of As that can be adsorbed at pH = 2 is 303 mg/g, which is more than any prior MOF composites that can remove As [143]. In recent years, some scholars have synthesized an environmentally friendly, economical, and efficient MIL−88B(Fe) [144]. It has practically all of the benefits of MOFs, including hydrophilicity, a flexible and programmable framework, and a porous structure. The primary As−oxygen bonding and coordination of FeO clusters inside the oxygen molecular framework constitute the adsorption process. The highest amount of As that may be absorbed is 156.7 mg/g.

ZIF−8 is a MOF material that can coremove Sb and As [145]. The results showed that the maximal As(III), As(V), and Sb(V) adsorption capabilities were attained at pH = 8.6, which were 151.3 mg/g, 106.4 mg/g, and 104.7 mg/g, respectively. The adsorption impact was lessened in the coexistence system because Sb and As engaged in competing for adsorption. However, this negative effect will disappear as the pH value and the initial As concentration rise. As a result, ZIF−8 can be thought of as a cutting−edge adsorbent that can remove Sb and As at the same time.

### 5.2. COFs Removal of Antimony and Arsenic

A brand−new variety of organized active crystal porous polymer is called a covalent organic framework (COF). There is no pertinent research on the adsorption of Sb by COFs, while the investigation of As adsorption by COFs is still in the exploratory stage with very few findings.

The earliest synthesized COF composite material that can adsorb As is γ−Fe_2_O_3_@CTF−1 [146]. For As(III) and As(V), its highest adsorption capabilities were 198.0 mg/g and 102.3 mg/g, respectively. Additionally, this sort of COF composite material has consistently maintained strong adsorption capability and may be recycled numerous times. Yang et al. successfully prepared an EB−COF composite material that is suitable for removing arsenate in water and the maximum As adsorption capacity reached 53.1 mg/g [72]. The main mechanism of its adsorption is the electrostatic interaction between the negative charge on the phosphate or arsenate ion and the positive charge on the COF (=N^+^—) and the interaction between the H atom on the phosphate or arsenate ion and the COF (—C=O) hydrogen bonding between groups. Liu et al. successfully prepared Fe^0^/TAPB−PDA COFs by an in situ growth method, and the adsorption process of As(III) is shown in Figure 6 [147]. The porous surface of COFs provided effective locations for both reaction sites and Fe^0^ loading. This COF composite performed well when compared to pure nZVI (nano zero−valent iron), and its adsorption capacity reached 135.78 mg/g.

### 5.3. Graphene to Remove Antimony and Arsenic

Graphene is a new two−dimensional honeycomb lattice material formed by the accumulation of sp^2^ carbon atoms [106]. In recent years, many scholars have studied the combination of different materials and graphene to further improve the Sb and As adsorption performance.

In a study by Leng et al., graphene was employed as an adsorbent to remove Sb(III) from water [148]. The results revealed that graphene had a maximum Sb(III) adsorption capacity of 10.9 mg/g. Then, several researchers demonstrated that graphene oxide (GO) had a strong adsorption capacity for Sb(III) in water, which was higher than that of many other adsorbents, including biosorbents, bentonite, graphene, and so forth [149]. A 3D nanostructured composite adsorbent of reduced graphene oxide and Mn_3_O_4_ was created by Zou et al. using a solvothermal reaction and reflux condensation combination [150]. For Sb(III) and Sb(V), their maximum adsorption capabilities were 151.8 mg/g and 105.5 mg/g, respectively. Similar to that, As was also removed using graphene and its composite materials. The electrostatic interaction between the positive charges on the adsorbent surface and the As(V) anion is the main mechanism by which this composite adsorbent adsorbs As(V). For As(III) and As(V), its maximum adsorption values were 180.3 mg/g and 172.1 mg/g, respectively [151]. In the latest research, some researchers successfully created graphene oxide−supported organo−montmorillonite composites (GO−OM) with a maximum adsorption capacity of 80.20 mg/g for As(V), which can be employed as a practical and sustainable adsorbent [152].

### 5.4. Other Adsorbents to Remove Antimony and Arsenic

In the most recent study, some researchers successfully developed graphene oxide−supported organo−montmorillonite composites (GO−OM), which may be used as a useful and sustainable adsorbent and had a maximum adsorption capacity of 80.20 mg/g for As(V) [68,69]. Dai et al. synthesized nanoscale zero−valent iron (nZVI) for the adsorption of Sb(III) and Sb(V) in an aqueous solution, and the findings demonstrated that a suitable dose of nZVI may eliminate all Sb within 90 min [66]. Fe−Mn binary oxide (FMBO), which was created by Xu et al. to remove Sb(III), had an adsorption rate that reached 81.3%, compared to MnO2′s greatest adsorption rate of just 62.3% [59]. The BET surface area of FMBO is 231 m^2^/g, which is double that of MnO_2_, explaining why its adsorption capacity is larger than that of MnO_2_ and FeOOH. Additionally, the oxidation reaction that happened when FMBO adsorbed Sb(III) encouraged the adsorption of Fe to Sb (V). Lin et al. synthesized γ−Fe_2_O_3_ nanoparticles for the removal of As(III) and As(V) in water [153]. According to the study, As(III) and As(V) adsorption capacities for As(III) and As(V) reached 74.83 mg/g and 105.25 mg/g, respectively, within the first 30 min of the adsorption process. As(III) and As(V) were discovered to attain the adsorption equilibrium condition within 30 min of the adsorption procedure. Investigating the mechanism revealed that the material’s hydroxyl (—OH) surface density affects the adsorption capacity [154]. In the latest study, Wang et al. used Fe−Cu binary oxides to coremove Sb(V) and As(V) from aqueous solutions and explored the coremoval mechanism in depth [155]. The adsorption amounts of ions reached 94.3 mg/g and 70.9 mg/g, respectively. This study offers a fresh concept for using iron−based bimetallic oxides in the removal of Sb and As from cores.

Due to its cross−linked polymer network, composite hydrogels contain a large number of hydrophilic groups and abundant functional groups and are easy to recycle, and are often used to adsorb heavy metals. Yuan et al. synthesized Fe−Mn binary oxide doped hydrogel (PPAA−FMBO_3_) to remove antimony in a water environment [75]. The adsorbent can efficiently bind Sb(III) up to 105.59 mg/g. The hydroxyl (−OH) and carboxyl (−COOH) functional groups in PPAA−FMBO_3_ serve as Sb(III) adsorption sites. This study demonstrates the advantages of hydrogel composites, such as strong adsorption capacities, adaptability to different pH values, no secondary pollution, and easy recovery and separation. It can be used as a potential composite frontier material for adsorbing Sb and As.

The research results of frontier materials on the adsorption of Sb and As are listed in Table 4. Compared with traditional adsorption materials, frontiers of adsorption materials show better adsorption performance. In comparison to bentonite, NU−1000 had an adsorption impact on Sb(III) and Sb(V), which was 244.6 and 575.8 times greater, respectively. As(III) and As(V) in Zn−MOF−74 performed adsorption five and six times better than iron−modified activated carbon, respectively.

## 6. Conclusions and Outlook

### 6.1. Conclusions

This article combines a large number of literature surveys and statistical analyses, and the conclusions are listed as follows.
In recent years, pollution incidents have occurred frequently, and the combined pollution of Sb and As is common. How to efficiently control combined pollution is one of the key areas of heavy metal pollution control.The current methods for removing Sb and As in water environments mainly include coagulation/flocculation, ion exchange, membrane technology, phytoremediation, and electrochemical methods. Compared with the above technologies, adsorption technology has high efficiency in removing Sb and As. Meanwhile, it has the advantages of low cost, high benefit, strong regeneration ability, no by−products, and simple operation.The type of adsorbents for heavy metals in sewage has changed from traditional adsorbent materials such as activated carbon and zeolite to the frontier of adsorbent materials with better adsorption effects, such as MOFs and COFs. Compared with traditional materials, these materials have a larger adsorption surface area, lower cost, and more flexible and adjustable structure.At present, the way of using micro carbon composite materials to treat heavy metals such as Sb and As has been accepted by more people due to the high efficiency of metal absorption ability. As a frontier adsorption material, COFs have been used to remove As in water environments with good adsorption effects. However, relevant research on Sb adsorption has not been carried out, and Sb is a kind of adsorption material with great potential.

### 6.2. Outlook

Based on the above conclusions, the current problems and the future development potential of frontier materials for the removal of Sb and As in water environments are summarized as follows.
The process of removing antimony and arsenic by various adsorbents is significantly affected by various factors, such as pH, initial concentration of antimony, arsenic in the solution, adsorbent dosage, and competitive ions. Future research on the removal of antimony and arsenic must not only overcome many unfavorable factors and improve the removal efficiency of antimony and arsenic but, more importantly, focus on developing new materials that are economical, environmentally friendly, and recyclable.Iron−based materials are highly efficient adsorption materials. Iron is thought to be the most effective metal at repairing antimony adsorption sites. It also appears to have some influence over arsenic adsorption. Additionally, it is simple to recycle and convert it into HFO, which can successfully encourage the adsorption of antimony and arsenic by the adsorbent. In particular, antimony has an adsorption impact that is many times greater than that of ordinary materials. Furthermore, iron−based MOFs can more effectively adsorb heavy metals in solution and use the coordination of coordination bonds with the formed HFO to accelerate the adsorption of antimony and arsenic in water. The materials are easier to recycle and reuse, thereby reducing costs and by−products. As a result, using iron−based materials to adsorb antimony and arsenic can significantly increase their adsorption capacity, making this a useful adsorption technique.Although many frontier materials, such as MXenes and HOFs, have not been used to study the adsorption of antimony and arsenic in water, they still have great research value. Among them, HOF materials are often used for gas adsorption, but their adsorption of metals is lacking. Given their similar structure to MOFs and COFs, the preparation is relatively simple, so HOFs have great application potential; MXenes have not been used to remove antimony and arsenic, but their structure is similar to graphene. MXenes have a larger surface area than graphene and are flexible and adjustable. COF materials are often used to remove heavy metals in water. Compared with MOFs, they have a more ordered channel structure, higher thermal and chemical stability, and lower density. Therefore, as a highly potent adsorbent, COFs have a significant effect on the adsorption of Cr, As, Hg, etc. However, COF materials have not been used to adsorb antimony. Given the strong antimony adsorption on iron−based metal−organic frameworks, iron−based covalent organic frameworks have a lot of potential for antimony adsorption research.Even though new adsorbents with outstanding performance are constantly being developed, recent research has discovered that these materials frequently struggle with poor desorption efficiency. After multiple cycles, the adsorption capacity decreases as a result of strong chemical interactions and redox conditions. There are few studies on how to improve the recycling rate of adsorbents and the disposal of waste adsorbents, which deserve further investigation.Currently, most adsorption studies are focused on simulating the adsorption performance of adsorbents in wastewater, including the exploration of adsorption isotherms, equilibrium, and adsorption kinetics. These are undoubtedly important, but practical methods for removing antimony and arsenic should also be actively explored in the research.In the current research on the removal of antimony and arsenic, many adsorption materials can remove antimony and arsenic alone, such as manganese dioxide, titanium dioxide, and nano zero−valent iron. However, there is a lack of research on the coremoval of antimony and arsenic. In future studies on the adsorption of antimony and arsenic, the adsorption performance of the adsorbent should be continuously improved. The combined removal of antimony and arsenic by a certain adsorbent can be compared with the single adsorption effect of antimony and arsenic. Exploring the feasibility of coremoval of antimony and arsenic will lay the foundation for in−depth research on removing antimony and arsenic.


## Figures and Tables

**Figure 1 ijerph-19-10824-f001:**
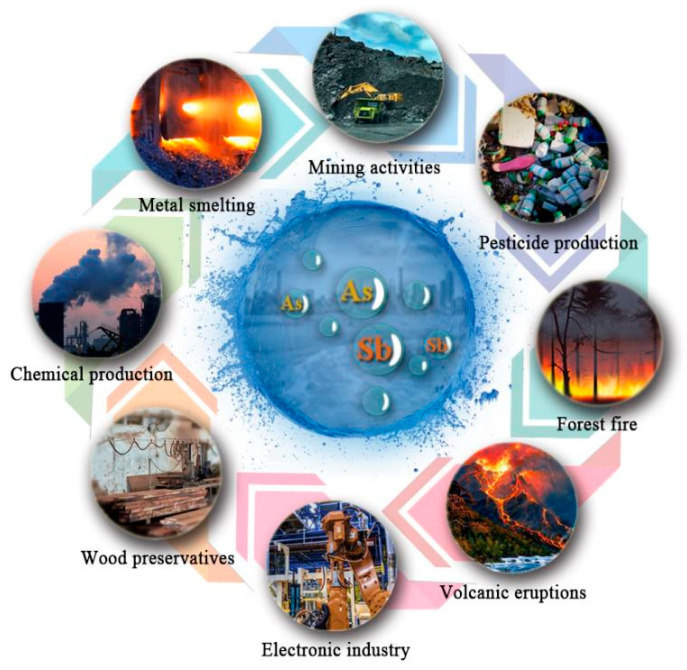
Sources of Sb and As pollution.

**Figure 2 ijerph-19-10824-f002:**
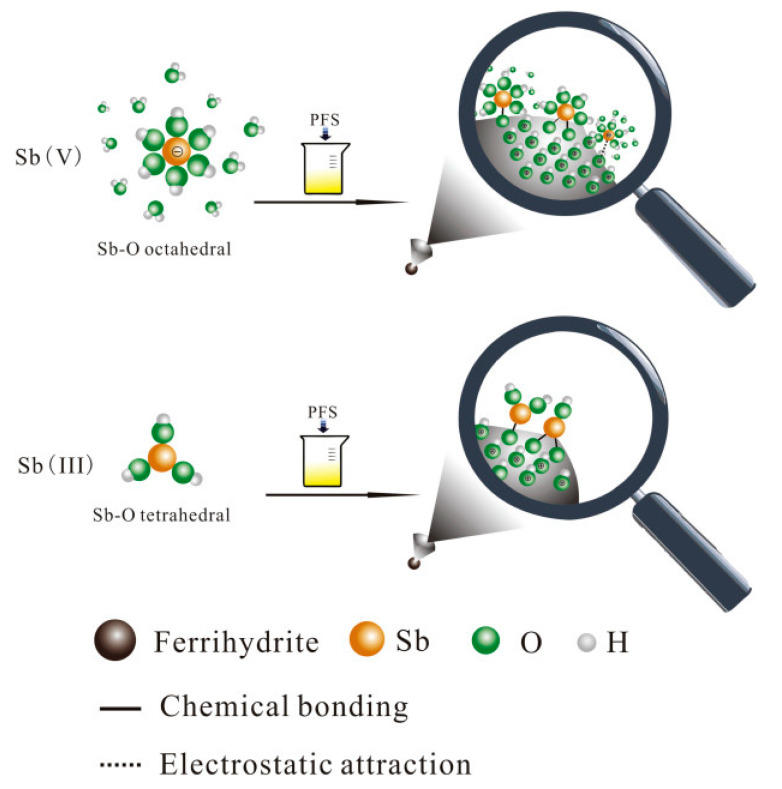
Sb(V) and Sb(III) removal mechanism in the CFS process [32]. Copyright 2018, Elsevier Publishing Group.

**Figure 3 ijerph-19-10824-f003:**
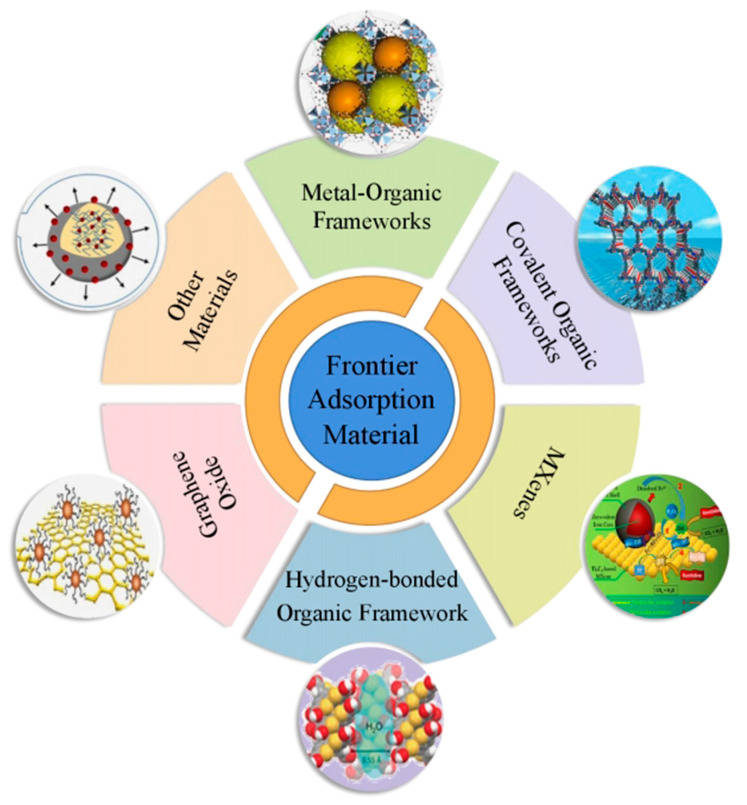
Frontier material classification map.

**Figure 4 ijerph-19-10824-f004:**
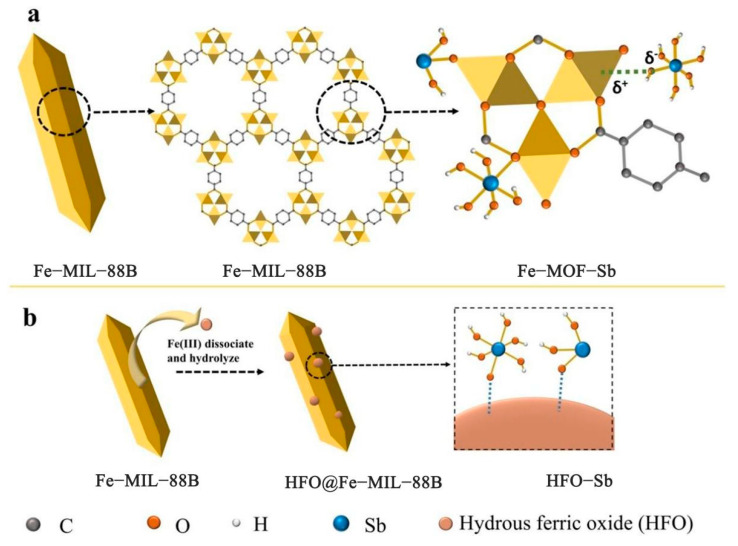
(**a**) Complexes Fe−O−Sb formed by incompletely coordinated Fe ions with Sb(III)/(V), (**b**) In situ formation of HFO (hydrated iron oxide) provides additional active sites for adsorption [140]. Copyright 2020, Elsevier Publishing Group.

**Figure 5 ijerph-19-10824-f005:**
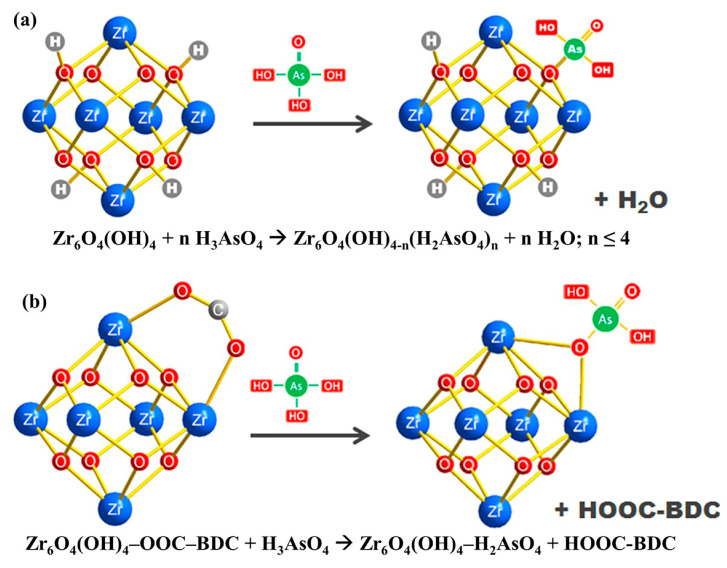
(**a**) Mechanism of arsenate adsorption by the hydroxyl group of UIO−66, (**b**) Mechanism of arsenate adsorption by the BDC ligand of UIO−66 [143]. Copyright 2015, Nature Publishing Group.

**Figure 6 ijerph-19-10824-f006:**
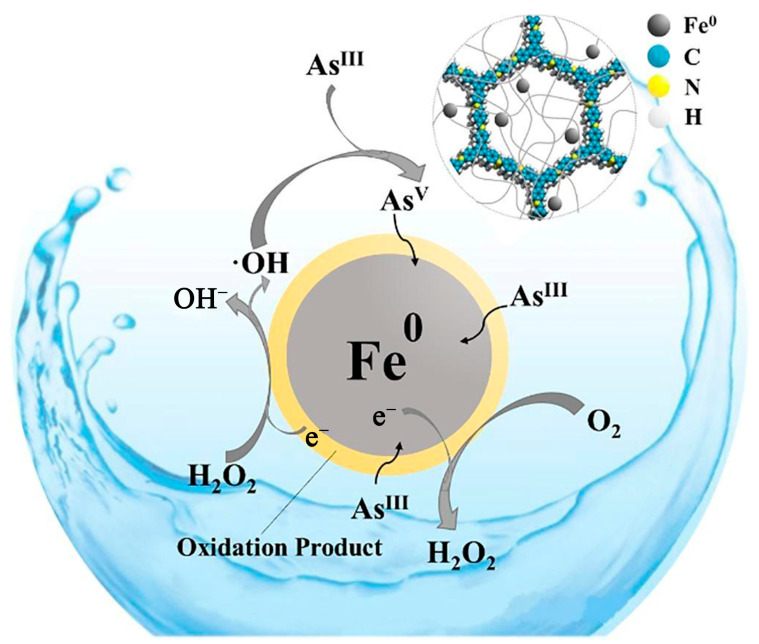
Schematic of proposed redox reactions on Fe^0^ surface during As(III) adsorption processes [147]. Copyright 2020, Elsevier Publishing Group.

**Table 1 ijerph-19-10824-t001:** Adsorption capacity and other parameters of different pollution treatment technologies for removing antimony and arsenic.

Method	Materials	Heavy Metal	Initial Concentration (mg/L)	Adsorption Temperature (°C)	Optimum pH	Adsorption Removal Efficiencies (%)	References
Coagulation/ flocculation	Ferric chloride	Sb(V)	0.05	25 ± 1	4.5–5.5	98	[31]
	HFO	Sb(III)/Sb(V)	0.1	25 ± 1	6	94/59	[31]
	Aluminum sulfate	As(V)	0.5	−	7	100	[47]
	Ferric chloride	As(III)/As(V)	1	−	7	60/90	[48]
Ion exchange	Purolite S957	Sb(III)	250	55	8	90	[49]
	Amberlite XAD−7	As(III)	−	−	5–10	>95	[36]
	Amberlite XAD−7	As(V)	−	−	1–5	>95	[36]
Membrane technology	CF−UF	Sb(III)	0.0625	28 ± 1	7.1–9.0	90	[50]
	NTR−729HF	Sb(III)	−	−	5	>60	[39]
	NTR−729HF	Sb(V)	−	−	3–10	>80	[39]
	ES−10	Sb(III)/Sb(V)	−	−	3–10	>80	[39]
	NTR−729HF	As(III)/As(V)	−	−	10	43/95	[39]
Electrochemical methods	Al−Al electrodes	Sb(V)	28.6	−	2	97.7	[51]
	Fe−Al electrodes	Sb(III)	0.521	−	5.2	99	[41]
	Copper–graphite	Sb	3500	−	−	99.4	[40]
	Zinc–zinc electrodes	As(III)	2	30	6	99.9	[52]
	Copper–copper electrodes	As(III)	2	30	7	99.6	[52]
	Stainless steel electrode	As	10	−	5.2	99.6	[53]
Phytoremediation	Cladophora	As	6	−	7.5	99.8	[54]

**Table 3 ijerph-19-10824-t003:** Comparison of the advantages and disadvantages of frontier and traditional adsorption materials.

Adsorption Materials		Advantages	Disadvantages	References
Traditional material	Carbon−based materials	Environmentally friendly Large surface area Thermal stability High adsorption capacity High mechanical strength Tunable surface functional groups	Difficult to recover/regenerate May cause secondary pollution Not suitable for industrial production	[123]
	Clay minerals	Low cost High surface area Selectivity and regenerability Excellent structural/surface properties Excellent physical/chemical properties	Low removal efficiency Easily affected by environmental factors	[82,124]
	Manganese oxides	Good stability Simple operation High surface area Porous structures Strong oxidation performance	High cost May cause secondary pollution Unfavourable physical/chemical properties	[125,126]
	Activated alumina	High surface area For commercial use High mechanical strength	High preparation cost Sensitive to pH	[127,128]
	Biomass material	Low cost Regenerative Environmentally friendly No secondary pollution Metal recovery possible	Effectiveness depends on pH and temperature Not suitable for industrial scale yet	[129,130]
Frontier materials	Metal−organic frameworks (MOFs)	High yields High surface area Multi−functionality Superior reusability Selective adsorption Good chemical/thermal stability Weakly toxic secondary products A highly ordered porous structure	Ions leakage Low conductivity Low−water stability	[131,132]
	Covalent organic frameworks (COFs)	Low density Great stability High adsorption capacity Large specific surface area Tunable, ordered, and stable structure	High cost Low yield Low crystallinity Poor reversibility Long synthesis time Complicated synthesis	[133,134]
	Hydrogen−bonded frameworks (HOFs)	Easy purification Good recyclability Solution processability Highly crystalline structure	Structural instability No outstanding porosity	[102]
	Graphene oxide (GO)	Low cost High mechanical strength Abundant functional groups Large specific surface area Strong adsorption capacity	Difficult to recycle and reuse Difficulty in large−scale synthesis Inactive surface chemical properties	[135]
	MXenes	Hydrophilicity Large surface area High sorption selectivity Remarkable chemical stability High thermal/electrical conductivity Exclusive sorption−reduction capacity	Low yield High cost Possibly poisonous Weak water stability	[136]
	Iron−based materials	Low cost Strong reactivity Easy preparation and recycling Great affinity towards Sb and As	Sensitive to pH Subject to corrosion Tendency to agglomerate Limited adsorption capacity Not thermodynamically stable	[69]
	Hydrogels (HGs)	Insoluble Non−toxic hydrophilicity Thermo−stability Controllable pore structure	High crystallinity Soluble in dilute acid Poor chemical resistance Poor mechanical strength Limited adsorption capacity	[137]

**Table 4 ijerph-19-10824-t004:** Adsorption capacity and other parameters of different frontier adsorbents for removing Sb and As.

Adsorbent		Heavy Metal	Initial Concentration(mg/L)	Adsorbent Dose (g/L)	Adsorption Temperature (°C)	Optimum pH	Adsorption Capacity (mg/g)	References
MOFs	ZIF−8	Sb(V)	0.06–1.1 mmol/L	0.2	25	8.6	104.7	[145]
	UIO−66−NH_2_	Sb(III)/Sb(V)	500	1	25	1.5	61.8/105.4	[139]
	Fe−MIL−88B	Sb(III)/Sb(V)	0.06–30	0.02	25	10/6	566.1/318.9	[140]
	NU−1000	Sb(III)/Sb(V)	2–500	0.8	−	11/3	137.0/287.9	[138]
	Fe−based MIL−88A	As(V)	100	0.4	25	5	145	[141]
	MIL−53 (Al)	As(V)	0.054–2.428	0.02	25	8	105.6	[142]
	MIL−88B (Fe)	As(V)	0.1–10	0.02	−	6	156.7	[144]
	UiO−66	As(V)	50	0.5	25 ± 1	2	303.4	[143]
	Cubic ZIF−8	As(III)	5–70	0.2	25 ± 0.5	8.5	122.6	[156]
	Leaf−shaped ZIF−8	As(III)	5–70	0.2	25 ± 0.5	8.5	108.1	[156]
	Dodecahedral ZIF−8	As(III)	5–70	0.2	25 ± 0.5	8.5	117.5	[156]
	ZIF−8	As(III)/As(V)	0.06–1.1 mmol/L	0.2	25	8.6	151.3/106.4	[145]
	Zn−MOF−74	As(III)/As(V)	800	1	25	12/7	211.0/325.0	[157]
COFs	γ−Fe_2_O_3_@CTF−1	As(III)/As(V)	10	4	−	7	198.0/102.3	[146]
	EB−COF: Br	As(V)	4	1	25	7	53.1	[72]
	Fe^0^/TAPB−PDA COFs	As(III)	173	0.17	−	8	135.8	[147]
Graphene	PAG	Sb(III)	1–25	1.5	20	5	158.2	[158]
	GO−SCH	Sb(V)	0–55	0.3	25 ± 1	7	158.6	[159]
	RGO/Mn_3_O_4_	Sb(III)/Sb(V)	10–1000	1	20	7	151.8/105.5	[150]
	CMGO	As(III)	10	5	25	7.3	45.0	[160]
	GO−OM	As(V)	0–250	1	−	7	80.2	[152]
	Fe_3_O_4_−HEG	As(III)/As(V)	50–300	−	−	−	180.3/172.1	[151]
	M−GO	As(III)/As(V)	0.15–1	1	25	7/4	85.0/38.0	[161]
Others	FMBO	Sb(III)	0.2–2 mmol/L	0.2	20 ± 1	3	203.3	[59]
	PPAA−FMBO_3_	Sb(III)	40	1	15	5	105.6	[75]
	γ−Fe_2_O_3_ nanoparticles	As(III)/As(V)	10–150/10–200	1.6	50	6/3	74.8/105.3	[153]
	Fe−Cu binary oxides−2/1	Sb(V)/As(V)	10–100	0.1	−	4	94.3/70.9	[155]

## Data Availability

Not applicable.

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
