# Peer review of "Frontier Materials for Adsorption of Antimony and Arsenic in Aqueous Environments: A Review"

_ijerph, 2022, doi:10.3390/ijerph191710824_

Round 1

Reviewer 1 Report

This article focuses on summarizing the characteristics, adsorption
mechanism, and performance of the frontier adsorption materials reported so far, describing their research and application progress in the removal of antimony and arsenic in the  water environment, and conducting an objective analysis and comparative evaluation of  the adsorption effects of a variety of frontier adsorption materials.

The review clear, comprehensive and of relevance to the field, a gap in knowledge identified. This review is divided into the following parts: (1) Chemical properties, hazards, and  source distribution of Sb and As; (2) Sb and As pollution control technology; (3) Characteristics, mechanism, and application of frontier adsorption materials; (4) Research progress on the adsorption of Sb and As by frontier materials; (5) Summary and prospects. This review combines a large number of literature surveys and statistical analyses,  the statements and conclusions drawn coherent and supported by the listed citations. The figures and  tables are appropriate, they properly show the data and easy to interpret and understand.

Author Response

Dear Reviewer:

Thank you for taking the time out of your busy schedule to read.

Reviewer 2 Report

1. P2 L49 minging activities or should it be mining activities?

2. P2 L77 you can show the dominant forms of As and Sb at various pH and how they change

3. When you discuss the presence of As or Sb in aqueous environment you should provide its ionic name

4. I would recommend to discuss the overall cost-effectiveness of the methods presented in the review. You can use the discussion from P7 L249-L258 in a sepaeare section and improve it by more detail analysis of the problem.

5. Table 1,2,4: please provide more details on the adsorption parameters: adsorbent dose, adsorption temperature, etc. Otherwise it gives no fair comparison of the process.

6. Section 4.please improve this section in terms of materials characterization in subsections - latest findings, synthesis of the materials, etc. I am also missing the presentation of biomass derived materials. Please revise.

7. Figure 3. This is my personal opinion but I would present this image without rotation to make it easier to read.

8. As the manuscript is a review article, the literature older than 3 years should not be used. Please provide the most latest results from the literature and make it more up to date. 

9. Please revise the work in terms of typos and grammar errors to improve its quality.

Author Response

Dear Reviewer:

First of all, thank you very much for taking your precious time out of your busy schedule to read this paper and provide objective and valuable opinions. The following are the adjustments we have made based on your opinions:

1.P2 L50  Changed minging activities to mining activities.

2.P2 L75-76 Add the influence of environmental factors, the original text is changed to Chemical properties, hazards, source distribution and forms of existence in different environments of Sb and As.

3.P4 L132-151The two factors (pH and redox conditions) that most affect the existence of Sb and As in the water environment and their ion names are described respectively. In addition,P4 L152-156 and L167-168 supplement its existence in different water environments (aerobic and anaerobic).

4.P9 L318-345 Supplementary subsection 3.7, which describes the advantages and limitations of various techniques.

5.Added more details to Tables 1, 2, 4.

6.Section 4 has supplemented the latest progress of various cutting-edge materials based on the latest research; biomass materials have been deleted, and are considered to be traditional materials after discussion.

7.Figure 3 Modifying the angle of the picture to make it easier for readers to read.

8.Update nearly 30 papers, because the papers in the past 3 years are limited and need a lot of foreshadowing of previous research (such as comparing traditional materials and new frontier materials).Most of the papers are revised to papers in the past 5 years.

9.Grammar and spelling errors in the article have been carefully revised.

Round 2

Reviewer 2 Report

The Authors have revised their work according to suggestions. I can therefore recommend their work to the Reviewer for publication.